# A Novel Alternative in the Treatment of Detrusor Overactivity? In Vivo Activity of O-1602, the Newly Synthesized Agonist of GPR55 and GPR18 Cannabinoid Receptors

**DOI:** 10.3390/molecules25061384

**Published:** 2020-03-18

**Authors:** Andrzej Wróbel, Aleksandra Szopa, Anna Serefko, Ewa Poleszak

**Affiliations:** 1Second Department of Gynecology, Medical University of Lublin, 8 Jaczewskiego St., PL-20954 Lublin, Poland; 2Chair and Department of Applied and Social Pharmacy, Laboratory of Preclinical Testing, Medical University of Lublin, 1 Chodźki St., PL-20093 Lublin, Poland; anna.serefko@umlub.pl (A.S.); ewa.poleszak@umlub.pl (E.P.)

**Keywords:** animal model of detrusor overactivity, GPR55 agonist, GPR18 agonist, O-1602, biomarkers of overactive bladder, rat

## Abstract

The aim of the research was to assess the impact of O-1602—novel GPR55 and GPR18 agonist—in the rat model of detrusor overactivity (DO). Additionally, its effect on the level of specific biomarkers was examined. To stimulate DO, 0.75% retinyl acetate (RA) was administered to female rats’ bladders. O-1602, at a single dose of 0.25 mg/kg, was injected intra-arterially during conscious cystometry. Furthermore, heart rate, blood pressure, and urine production were monitored for 24 h, and the impact of O-1602 on the levels of specific biomarkers was evaluated. An exposure of the urothelium to RA changed cystometric parameters and enhanced the biomarker levels. O-1602 did not affect any of the examined cystometric parameters or levels of biomarkers in control rats. However, the O-1602 injection into animals with RA-induced DO ameliorated the symptoms of DO and caused a reversal in the described changes in the concentration of CGRP, OCT_3_, BDNF, and NGF to the levels observed in the control, while the values of ERK1/2 and VAChT were significantly lowered compared with the RA-induced DO group, but were still statistically higher than in the control. O-1602 can improve DO, and may serve as a promising novel substance for the pharmacotherapy of bladder diseases.

## 1. Introduction

According to the International Continence Society definition, overactive bladder (OAB) is a chronic ailment with urinary urgency, and usually presents with enhanced daytime frequency and nocturia as the main symptoms. In addition, urinary incontinence is observed in some patients [1,2,3,4]. OAB, one of the most prevalent lower urinary tract (LUT) conditions, is diagnosed as often in females as in males (in total approximately 10–17% of the population over 18 years of age) [5,6,7], but gender-dependent differences in manifestations, mainly caused by anatomical and physiological dissimilarities in the lower part of the urinary tract, are observed [8,9,10,11]. OAB has a great influence on the quality of life of millions of people worldwide. It is well established that OAB symptoms significantly lower the quality of life, mainly via its impact on everyday activities [3]. Additionally, people suffering from OAB very often experience confusion and frustration, which contributes to the exacerbation of the stress and the occurrence of depressive disorders and anxiety, causing withdrawal from society [7,12,13,14].

Most pharmaceuticals currently prescribed for patients with OAB have a peripheral effect. The golden standard in OAB therapy is muscarinic receptor antagonists [15]. These drugs work primarily by inhibiting muscarinic M_3_ receptors, which are located at the neuromuscular junctions in the detrusor muscle. They mainly reduce the contractility of the detrusor muscle and affect the bladder’s sensory nerves [15,16,17]. Antimuscarinics mitigate the severity of OAB symptoms (e.g., reduce pressure on the bladder, stabilize detrusor overactivity (DO), increase bladder volume, etc.). However, their clinical usefulness is limited by well-known and often occurring adverse effects, such as inter alia dry mouth, blurred vision, constipation, drowsiness, cognitive impairment, and not always satisfactory clinical effects [16,17]. Furthermore, to achieve their maximum therapeutic effect, muscarinic receptor antagonists require a regular administration for several weeks at a minimum [18]. These curtailments of the most commonly prescribed drugs force intensive search for new strategies for the treatment of OAB.

Pre-clinical studies conducted in the recent years have led to the discovery of new potential targets for OAB pharmacotherapy, such as Rho-kinase (ROCK) [19,20], nerve-growth factor (NGF) [21,22], K^+^ channels (ATP-sensitive channels—K_ATP_, and calcium channels—K_Ca_) [23], serotoninergic (5-HT_1A_, 5-HT_2_, 5-HT_3_, and 5-HT_4_) [24], dopaminergic (D_1_–D_5_) [25,26], β_3_-adrenergic [27,28,29,30], vitamin D_3_ [31,32], kinin (B1 and B2) [33], oestrogen [34], opioid (μ, δ, and κ) [35,36], purinergic receptors (P_2_X_1_ and P_2_X_3_), transient receptor potential vanilloid 1 (TRPV_1_) and 4 (TRPV_4_) [37,38], tachykinin (NK1 and NK2) [39], glutamatergic (NMDA and AMPA) [40], γ-aminobutyric acid (GABA_A_ and GABA_B_) [41,42], corticotropin-releasing factor (CRF) [43], and cannabinoid receptors (CB1, CB2, and GPR55) [44,45,46]. All these possibilities of pharmacotherapy for DO/OAB have limitations. The most common limitations in their use are side effects, especially central, occurring during the therapy and the fact that their effectiveness appears with a delay of several weeks [19,20,21,22,23,24,25,26,27,28,29,30,31,32,33,34,35,36,37,38,39,40,41,42,43,44,45,46]. The activity of most of them has not yet been confirmed in clinical assessments. Nowadays, clinical studies are being conducted only for β_3_-adrenergicreceptor agonists, K^+^ channels opening compounds, and ROCK inhibitors [27].

In 1997, Consroe et al. [47] published the outcomes of a research which indicated the beneficial impact of marijuana on bladder dysfunction (in particular, urinary urgency was improved by 64%, urgency incontinence by 55%, and hesitancy in initiating micturition by 59%) in patients with multiple sclerosis, which prompted a number of experiments aimed at determining the role of the endocannabinoid system in the pathophysiology and therapy of OAB [45]. In the literature, there are several reports on the impact of the endocannabinoid system on bladder function in laboratory animals and humans [48,49]. Several clinical trials have indicated that oral agents which modulate cannabinoid receptor activity might be an alternative therapy for patients with OAB resistant to antimuscarinic drugs [46,50,51]. Cannabinoids, endocannabinoids, and synthetic cannabinoid analogues exert their activity through the agonism of cannabinoid receptors, i.e., CB_1_, CB_2_, and G-protein-coupled receptor 55 (GPR55). Two of these, namely CB_1_ and CB_2_, are well-known and have been well-described, while GPR55 was relatively recently identified and not yet wellstudied [52,53,54]. In all layers of the bladder, both CB_1_ and CB_2_ receptors were identified (the expression of these receptors is higher in the urothelium than in the bladder, and the density of CB_1_ is greater than that of CB_2_ receptors) [44,45,55]. The latter ones were also identified in detrusor cholinergic neurons and sensory suburothelial nerves expressing TRPV_1_ and the calcitonin gene-associated peptide (CGRP) [45,55]. Furthermore, functional studies conducted by Bakali et al. (2013) [48] showed that CB_2_ receptors exerted both pre- and post-synaptic inhibitory effects on the contraction of the rat bladder, while CB_2_ receptors exclusively act post-synaptically [49]. Stimulating both CB_1_ andCB_2_ receptors inhibits adenylyl cyclase from generating the cyclic-adenosine monophosphate (cAMP) by interaction with the α-subunit of the G protein of the G_i/o_ family [56]. In turn, the orphan metabotropic GPR55 receptors (sometimes called CB_3_ receptors) [57], among others, are also localized in the bladder [58], mainly in the urothelium [59]. GPR55 receptors have an impact on a number of pathways, including the release of Ca^2+^ from intracellular storages and ROCK-dependent signaling, the nuclear factor of the activated T cell (NFAT), κ nuclear factor B (NF-κB), and cAMP response element binding (CREB), through activating the G_q_ and G_α12/13_ proteins [60].

There are few reports on the possibility of using substances that modulate cannabinoid receptor activity in the therapy of patients suffering from OAB. In the literature neither pre-clinical nor clinical studies that evaluated the possibility of using O-1602 in patients with DO were found. Therefore, the objective of the current research was to evaluate the effects of O-1602, an atypical synthetic cannabinoid which acts as a potent agonist of the GPR55receptor and biased agonist of the GPR18 receptor [61,62], on urodynamic parameters in the animal model of DO induced with the transient intravesical infusion of retinyl acetate (RA), in conscious female rats during cystometry. At the same time, the aim was to evaluate whether O-1602 affects the level of OAB-specific biomarkers. It should be emphasized that these are the first studies aimed at determining the possibility of using O-1602 in OAB therapy.

## 2. Results

### 2.1. Cystometry

The effects of RA, O-1602, and the joint administration of these agents on cystometric parameters are presented in Table 1.

A 5-min exposure of the urothelium to a 0.75% RA solution led to changes in cystometric parameters commonly considered to be typical in OAB. A significant enhancement was noted for the non-voiding contractions amplitude (ANVC, cmH_2_O), the area under the pressure curve (AUC, cmH_2_O/sec), the basal pressure (BP, cmH_2_O), the detrusor overactivity index (DOI, cmH_2_O/mL), and the non-voiding-contractions frequency (FNVC, times/filling phase), whereas the bladder compliance (BC, mL/cmH_2_O), the inter-contraction interval (ICI, s), the threshold pressure (TP, cmH_2_O), the volume threshold (VT, mL), the volume threshold to elicit NVC (VTNVC, %), and the voided volume (VV, mL) were found to decrease significantly. The instillation of the female rat bladder with RA proved not to affect the bladder contraction duration (BCD, s), the micturition voiding pressure (MVP, cmH_2_O), the post-void residual (PVR, mL), the relaxation time (RT, s) or the voiding efficiency (VE, %).

The administration of O-1602 to saline pretreated rats did not significantly affect any of the examined cystometric parameters. However, injecting this agonist into female rats with RA-induced DO ameliorated the symptoms of OAB, as evidenced by the increase in BC, ICI, TP, VT, VTNVC, and VV, and the decreases in ANVC, AUC, BP, DOI, and FNVC. O-1602 used in the RA-treated female rats was found not to have a statistically significant impact on BCD, MVP, PVR, RT, or VE. Importantly, following the administration of O-1602 to animals with RA-induced DO, the ANVC, AUC, BC, BP, FNVC, ICI, TP, VT, VTNVC, and VV values returned to the levels observed in the control group (CON).

### 2.2. Diuresis and Cardiovascular Parameters

The effects of RA and O-1602, and the joint administration of these agents, on the diuresis and cardiovascular parameters, are presented in Table 2. RA, O-1602, and co-administration of these agents did not significantly affect the urine production (UP, mL/day), the heart rate (HR, beats/min), or the mean arterial pressure (MAP, mmHg), as compared with the CON.

### 2.3. Biochemical Studies

The effects of RA, O-1602, and the joint administration of these agents on the level of the calcitonin gene-related peptide (CGRP), rat phosphorylation extracellular signal-regulated kinase 1/2 (ERK1/2), organic cation transporter 3 (OCT_3_), vesicular acetylcholine transporter (VAChT), brain-derived neurotrophic factor (BDNF), and nerve growth factor (NGF) are presented in Figure 1 and in Table 3. Instillation of the bladder with RA led to an elevated level of CGRP, ERK1/2, and OCT_3_ in the urothelium, of VAChT in the bladder detrusor, and of BDNF and NGF in the urine. O-1602 was found not to affect the above biomarkers in saline pretreated female rats. In turn, the administration of O-1602 to RA-treated female rats caused a reversal of the described changes, including in the case of CGRP, OCT_3_, BDNF, and NGF, to the levels observed in the CON. The values of ERK1/2 and VAChT were significantly lowered compared with those for the animals with RA-induced DO, but were still statistically higher than in the CON.

## 3. Discussion

To the authors’ best knowledge, this is the first study in which the effect of the O-1602 receptor agonist was assessed on the bladder function in an animal model of RA-induced DO. The aim of the study was to assess in vivo the effect of O-1602 on the micturition cycle of conscious animals, female rats, in the animal model of DO [63] previously developed by the research team, and its applicability in DO/OAB therapy. Previously, no attempt had been made to determine the level of the OAB biomarkers after the administration of these agents. The results of cystometric and biochemical studies presented in this manuscript support our hypothesis.

### 3.1. Cystometry

The results confirmed that bladder instillation with 0.75% RA induces changes in the cystometric parameters observed in the urodynamic incidence in patients with OAB [64]. In turn, the use of the GPR55 and GPR18 receptor agonist has been shown to improve DO symptoms in this animal model—O-1602 inhibited excessive bladder contractility in vivo. The impact of the tested compound on the micturition cycle was illustrated by performing a complete cystometry, which is the most reliable test performed in the diagnosis of OAB in humans. As mentioned above, the GPR55 receptors are localized in the bladder tissues and in the central nervous system (CNS) areas responsible for controlling urination [58,59]. While GPR18 is reported to be of highest level in the spinal cord and small intestine, with lower levels in the testis and cerebellum [62], which are regions that are not responsible for micturition. Therefore, the reported detrusor relaxation is probably resulting from GPR55 receptor stimulation. However, the participation of GPR18 receptors in the changes in the parameters studied cannot be excluded. This is consistent with the observations of other authors who used endo- and exogenous cannabinoids, which act via CB_1_ and/or CB_2_ receptors [44,55,65,66,67,68,69,70,71] localized in the LUT, and in the CNS regions, which are involved in micturition regulation [72].

Changes in cystometric parameters recorded in female rats with RA-induced DO treated with O-1602, i.e., an increase in BC, BCD, ICI, TP, VT, VV, and VTNVC, and a decrease in ANVC, AUC, BP, DOI, and FNVC, indicate that it may become an interesting alternative in drug therapy, both for OAB-wet and OAB-dry. In the current study, the reduction in AUC and BP in RA-induced OAB female rats treated with O-1602 to levels measured in the control group suggests that cannabinoid receptor stimulation can counteract the overall excitability of the bladders without affecting the maximum micturition voiding pressures [73]. Importantly, the lack of O-1602 impact on MVP, PVR, RT, and VE indicates that the applied dose only affects the urine accumulation phase, without affecting the voiding phase [74,75]. Notably satisfactory, is the observed decrease in the value of DOI—a parameter which seems to more accurately describe the degree of DO than other cystometric parameters [76], due to the fact that DO is a frequent manifestation of OAB. According to the literature data, there is a close correlation between DO and OAB, i.e., about 64% of patients with OAB have DO and 83% of patients with DO have OAB [77]. Additionally, the results of the authors’ study showed that the stimulation of the newly synthesized cannabinoid receptors GPR55 and GPR18 leads to a decrease in ICI, which is consistent with the results of other researches, which have shown that CB agonists reduce micturition frequency (MF), but increase VV (for review see [72]).

A very important outcome of the conducted experiment is the decrease in DOI following the administration of O-1602. This parameter is also used to assess the severity of DO in humans. Its evaluation consists of measuring the amplitude of the pressure curve during the conscious cytometry [78]. DOI depicts detrusor activity during the storage phase of the micturition cycle, and, therefore, more precisely characterizes myogenic detrusor activity if compared with ICI, BP, ANVC, FNVC, MVP, or BC [79]. The decrease in this parameter following the use of O-1602 indicates that this compound enhances the sensory threshold of the urinary bladder [80].

The rise of VTNVC (i.e., the pre-clinical counterpart of the urine volume excreted at the first involuntary detrusor contraction measured during urodynamic examinations in humans) [81], observed after the administration of O-1602 in female rats with DO induced by RA, is particularly interesting. Such changes in this parameter might be an evidence of the participation of GPR55 receptors in the control of bladder sensation, and can have important practical implications, as VTNVC is considered in clinical practice as a very reliable indicator of the effectiveness of OAB therapy, and its benefit is the effect of reducing micturition frequency and the number of urinary incontinence episodes [82]. However, the lack of any effect of the tested agent on cystometric parameters such as MVP, PVR, RT, and VE proves that it does not produce any impairment of the voiding function. The lack of any impact on RT might also indicate that O-1602 does not have any relaxing effect on the urethra, and, as a consequence, does not induce changes in outflow resistance.

Some published data suggest that afferent pathways are involved in the regulation of the micturition cycle, and further point to the role of cannabinoid receptors in modulating this activity (for review, see [72]). Hayn et al. [65] concluded that the inhibitory activity of the CB_1_/CB_2_ agonist—ajulemic acid—on rat bladder sensory activity observed in their experiments occurs via both these CB receptor subtypes. Furthermore, Walczak and co-workers [67] showed that the excitation of CB_1_ receptors expressed in the murine bladder after intravesical injection of the AZ12646915—a cannabinoid receptor agonist, decreased the bladder’s afferent fibers responsiveness by 40%, while pretreatment with AM251—a CB_1_ receptor antagonist—eliminated this effect. On that basis, they summarized that CB_1_ receptors are involved in the peripheral modulation of bladder afferent information. In subsequent experiments, Walczak and Cervero [70] also showed that the intravesical application of AZ12646915 significantly reduced the enhanced activity of fibrils in inflamed female mice bladders. As in previous research, this effect was blocked by CB_1_, but not the CB_2_ antagonist (AM251 and AM630, respectively). Similarly, clinical studies’ outcomes, although only indirectly, indicated the participation of CB_1_ and CB_2_ receptors in the adjustment of the human urinary tract afferent function [83]. The decrease in the value of cystometric parameters such as FNVC, ANVC, or DOI in the authors’ studies proves that GPR55 and GPR18 receptor stimulation can reduce the amount of frequency and urgency episodes, but also confirms the participation of this receptor in the afferent mechanisms which regulate the micturition cycle. The results obtained by the authors’ team show that not only CB_1_ and CB_2_ receptors, but also the orphan receptors GPR55 and GPR18, are involved in the afferent control of LUT.

### 3.2. Diuresis and Cardiovascular Parameters

GPR55 receptors in large quantities are located in the endothelium of blood vessels [84]. In the literature, there are some reports in which the effect of the GPR55 receptor agonist—L-alpha-lysophosphatidylinositol (LPI)—on arteries and arterioles, including mesenteric and pulmonary vessels, were examined. However, the outcomes of these researches are not conclusive. In 1996, Fukao et al. [85] noted that the use of LPI contributed to the elimination of acetylcholine-induced mesenteric artery relaxation in rats. In turn, AlSuleimani and Hiley indicated that LPI caused relaxation in the isolated rats’ small mesenteric arteries, and reduced BP in anesthetized rats [86]. Agonists of GPR55 receptors have also been shown to be able to induce the relaxation of isolated human pulmonary arteries [87]. These results imply that the agonist of GPR55 receptors can elicit either vasoconstriction or vasorelaxation. Recently, LPI has also been shown to affect the activity of rat neonatal ventricular cardiomyocytes in a GPR-dependent manner. It increases the intracellular concentration of calcium ions in cardiomyocytes by enhancing membrane hyperpolarization, and, consequently, causes myocardial constriction [88]. Due to the inconsistent reports on the impact of GPR55 receptor stimulation on cardiovascular parameters, one of the aims of the current study was to assess the O-1602 impact on HR, MAP, and UP. O-1602, at the applied dose (0.25 mg/kg, intra-arterially), did not alter the mentioned parameters in a statistically significant way. This might be important, especially in the context of the frequent coexistence of OAB and cardiovascular diseases in the elderly [88,89,90,91,92,93].

### 3.3. Biochemical Studies

To date, several potential OAB biomarkers have been identified in both pre-clinical and clinical studies, including CGRP, ERK1/2, OTC3, VAChT, BDNF, and NGF [22,72,94,95,96,97,98]. In this regard, the authors aimed to establish whether the administration of O-1602 is able to eradicate RA-induced disturbances in the levels of OAB markers in the bladder urothelium, detrusor muscle, and/or urine in female rats.

CGRP—an excitatory sensory neurotransmitter—is one of the bladder’s afferent activity markers [99]. Sensory afferent fibers in the bladder are structures in the bladder which contain large amounts of CGRP, from which it is released following chemical stimulation with capsaicin [100]. In pre-clinical experiments with the use of the cyclophosphamide (CYP)-induced cystitis model, a relationship between OAB and the level of CGRP was detected [99]. In this study, Vizzard [99] noted an enhancement in the CGRP level in rats with OAB. Furthermore, it was shown that, in an isolated rat bladder model, the evoked release of CGRP from afferent terminal nerves was reduced after the administration of the ajulemic acid—an endogenous mixed CB_1_/CB_2_ receptor agonist [55,101]. In the authors’ experiments, O-1602 was found to reduce the CGRP levels in animals with RA-induced DO without affecting the activity of afferent fibers in the controls. CB_2_ receptors are characterized by a high expression on sensory nerves, urothelium, suburothelium, and, in lower concentration, occur in the detrusor (mainly on cholinergic fibers), where they are localized in the vicinity of the TRPV_1_, CGRP, and VAChT (for review see [55,72]). RA stimulates activity in TRPV_1_ receptors [63], which might explain the increase in the CGRP and VAChT following their administration observed in the presented studies. In turn, the use of O-1602 significantly reduced the level of the CGRP, which returned to the value observed in the control group. The VAChT level also decreased, but it was still significantly higher than in normal animals. Possibly, the dose of O-1602 used in the presented research was insufficient to reduce it to the level noted in the control female rats. The increase in the VAChT concentration was congruent with the increase in acetylcholine transmission, which is the main neurotransmitter involved in detrusor contraction [26]. RA bladder instillation generates growth in the VAChT concentration in detrusor muscles and enhances its contractile activity, which is confirmed by the results of the cystometric examination, among others, the observed reduction in ICI. It follows that O-1602, by reducing the VAChT level, imparts a relaxing impact on the bladder muscles. Bladder urothelial cells, in contrast to neuronal cells, in which a high VAChT level is observed, are characterized by substantial OTC_3_ expression. According to the literature data, both these transporters participate in the release of acetylcholine from neuronal and non-neuronal cells, respectively [102,103,104,105]. Cholinergic receptors (nicotinic and muscarinic) are widely localized on non-neuronal cells, inter alia on urinary bladder epithelial cells [102,106,107]. Several authors emphasize that cholinergic mechanisms activated in the bladder by acetylcholine coming from non-neuronal tissues can play an important role in the initiation of LUT disorders, such as OAB [102,106,108], and can be a novel target for the treatment of these diseases. Therefore, the O-1602 agonist used in the authors’ study, whose use has contributed to reducing the amount of OCT_3_ and VAChT compared with the RA-treated group, might become an alternative to anti-muscarinic drugs commonly used in the therapy of patients suffering from OAB.

ERK1/2, a mitogen-activated protein kinases (MAPK) subfamily, has been shown to contribute a variety of changes in both neuronal and non-neuronal cells [109]. Additionally, it plays an important role in the control of the detrusor muscle tone and bladder reflex activity in the animal model of chronic bladder infection [110]. As demonstrated by Marentette et al. [111], ERK1/2 activation is greater in urothelial cells from the inflamed human bladders than in cells from normal bladders. In turn, Cruz et al. [110] showed that the application of a specific ERK phosphorylation inhibitor (PD98059) improves inflammatory-induced OAB. In the presented study, the authors found a significantly greater level of ERK1/2 in the urothelium isolated from rat bladders instilled with 0.75% RA, when compared with those measured in cells coming from the bladders of the control animals. To the authors’ knowledge, these are the first outcomes indicating that ERK up-regulation occurs not only in OAB induced by inflammation, but also in OAB triggered by the irritant stimulus, which in this case is RA, caused by the sensitization of nociceptive pathways [63,112]. Furthermore, in the literature, there was no information about the impact of cannabinoid receptor (either CB_1_, CB_2_, or GPR55) stimulation/inhibition on ERK1/2 signalization in OAB/DO. According to the authors’ research, O-1602 has the potential to reduce the ERK1/2 level increased in the applied animal model of DO. However, after using O-1602 at a dose of 0.25 mg/kg, the authors observed a significant decrease in the level of ERK1/2 in the bladder urothelium, but it did not return to the levels recorded in the normal female rats.

In the last stage of the biochemical research, the authors assessed the level of well-known and well-studied OAB biomarkers, i.e., NGF and BDNF, in the urine. Both of them belong to neurotrophins, which are essential, among others, for the survival and maintenance of neuronal cells located both peripherally and centrally [113]. Furthermore, these factors are produced in the bladder’s urothelial and smooth muscle cells, which are responsible for the modulation of peptidergic bladder sensory afferents [114]. High levels of NGF and BDNF are observed in both pre-clinical and clinical research in subjects with OAB, bladder pain, and/or cystitis syndromes [115,116,117,118,119,120,121]. Similarly, in the authors’ experiments, a significant increase in the concentration of both these neurotrophins in the applied RA-induced DO animal model was noted. In turn, a reduction in the NGF and BDNF levels was observed in patients suffering from OAB after the implementation of effective pharmacotherapy with antimuscarinic drugs [118,120,122,123,124] or botulinum toxin-A [122,125]. Also, intravenous and intrathecal injections of BDNF-antibodies significantly decreased the symptoms of OAB induced by cyclophosphamide in animals [126,127]. Jaggar et al. [128] demonstrated that the administration of endogenous CB_1_ as well as the CB_2_ receptor agonist (anandamide and palmitoylethanolamide, respectively) caused an NGF-dependent reduction in bladder hyperreflexion in the turpentine-induced bladder inflammation model. In addition, CB_1_ and CB_2_ receptors activation has been shown to reduce NGF-induced thermal and visceral hyperalgesia, while the use of CB antagonists potentiates it [129,130,131]. Intra-arterial administration of O-1602 in the authors’ research also contributed to decreases in the elevated levels of the NGF and BDNF in rats with RA-induced DO to the amount noted for normal female rats.

## 4. Materials and Methods

### 4.1. Experimental Animals

All the procedures performed in the studies involving animals were in accordance with the European Communities Council Directive of 22 September 2010 (2010/63/EU) and the Polish legislative Acts concerning animal experimentations. The experimental procedures and protocols were approved by the First Local Ethics Committee at the Medical University of Lublin.

All experiments were carried out on naïve female Wistar rats weighing 200–225 g purchased from a licensed breeder (The Experimental Medicine Centre, Lublin, Poland). The animals were placed individually in metabolic cages (3700M071, Tecniplast, West Chester, PA, USA) located in environmentally controlled rooms (temperature maintained at 21 ± 1 °C and humidity at 45–55%) with a 12-h light/dark cycle (lights on at 8:00 a.m.). Throughout the research, the rats were given unlimited access to standard rodent chow and water. The procedures were performed between 8:00 and 13:00 to minimize circadian influences.

### 4.2. Drugs

RA and O-1602 (5-Methyl-4-[(1*R*,6*R*)-3-methyl-6-(1-cyclohexen-1-yl]-1,3-benzenediol) were purchased from Sigma-Aldrich (Poznań, Poland) and Tocris (Bristol, Great Britain), respectively. RA (US Pharmacopeia Reference Standard, Fluka) was diluted to 0.75% (the minimum effective solution) with a mixture of Polisorbate 80 and saline, and administered to the bladder in the form of intravesical instillation for DO induction. Separately, O-1602 was dissolved in methyl acetate and administered intra-arterially via a polyethylene catheter inserted into the carotid artery during conscious cystometry in a single dose (0.25 mg/kg). The doses of the tested agents were selected on the basis of the outcomes of the authors’ previous research and the literature data, and were confirmed/adjusted in their laboratory in preliminary studies [59,63].

### 4.3. Treatment Schedule

The tested rats (*n* = 60) were randomly assigned to the following four experimental groups of 15 animals each:1^st^ group received saline (the control group, CON)2^nd^ group received RA3^rd^ group received O-1602 (0.25 mg/kg)4^th^ group received RA and O-1602 (0.25 mg/kg)

### 4.4. Surgical Procedures

To perform the surgery procedures, the animals’ general anesthesia was induced by using intraperitoneal injections of xylazine (15 mg/kg, Sedazin, Biowet) and ketamine hydrochloride (75 mg/kg, Ketanest, Pfizer) [30,63], which do not affect micturition in female rats [76]. The anesthetized animals were subjected to catheterization of the urinary bladder from the external urethral orifice with a polyethylene catheter. Subsequently, the residual urine was removed from the bladder and 0.75% RA (to induce DO) or saline was administered for 5 min using an intravesical instillation. After emptying the bladder from the RA solution/saline, its interior was gently washed with saline and the catheter was removed. Next, through a vertical midline incision (approximately 10 mm) the abdominal wall was opened and a double lumen catheter was inserted via the bladder apex (dome) and fixed with 6–0 absorbable suture. In the same surgical session, in order to administer the examined drug and vehicle directly into the vascular bed, and to measure the cardiovascular system parameters (i.e., heart rate and mean blood pressure), the carotid artery was cannulated [30,63,74,132].

To prevent infection, each rat was injected subcutaneously with 100 mg of cefazoline sodium hydrate (Biofazolin, Sandoz, Holzkirchen, Germany).

### 4.5. Cystometry

Three days after the surgical procedure, cystometric studies in conscious rats were carried out. A bladder catheter was connected via a three-way stopcock to the microinjection pump (CMA 100, CMA Microdialysis AB, Kista, Sweden) and also a pressure transducer (FT03, Grass Technologies, West Warwick, RI, USA), and then, the cystometry was carried out using saline at a constant rate of 0.05 mL/min to elicit repetitive voiding. A force displacement transducer (FT03C, Grass Technologies, West Warwick, RI, USA) linked to a fluid collector was used to measure micturition volumes. Additionally, cystometry profiles (cystometrograms) were recorded continuously with a Grass polygraph (Model 7E, Grass Technologies, West Warwick, RI, USA) [30,133]. During the cystometry procedure, the following parameters were evaluated: the area under the pressure curve (AUC, cmH_2_O/sec), the basal pressure (BP, cmH_2_O), the bladder compliance (BC, mL/cmH_2_O), the bladder contraction duration (BCD, s), the inter-contraction interval (ICI, s), the micturition voiding pressure (MVP, cmH_2_O), the post-void residual (PVR, mL), the relaxation time (RT, s), the threshold pressure (TP, cmH_2_O), the voided volume (VV, mL), the voiding efficiency (VE, %), the volume threshold (VT, mL), the detrusor overactivity index (DOI, cmH_2_O/mL), the non-voiding contractions amplitude (ANVC, cmH_2_O), the non-voiding-contractions frequency (FNVC, times/filling phase), and the volume threshold to elicit NVC (VTNVC, %) [30,74,79,132].

### 4.6. Diuresis and Cardiovascular Parameters

After the cystometric study, in order to evaluate the effects of the O-1602 on urine production (UP, mL/day), the heart rate (HR, beats/min), and the mean arterial pressure (MAP, mmHg), each animal was placed individually in a metabolic cage (3700M071, Tecniplast, West Chester, PA, USA) for 24 h.

### 4.7. Biochemical Studies

In the bladder urothelium, the levels of the following biomarkers were estimated: calcitonin gene-related peptide (CGRP; Biomatik, CN EKU02858), organic cation transporter 3 (OCT_3_, Antibodies-online, CN ABIN6227163), and rat phosphorylation extracellular signal-regulated kinase 1/2 (ERK1/2, Creative Diagnostics, CN DEIA-BJ2227). Additionally, in the bladder detrusor muscle, the levels of vesicular acetylcholine transporter (VAChT, LifeSpan BioSciences, CN LS-F12924-1) were estimated, in the urine, the concentration of nerve growth factor (NGF, LifeSpan BioSciences, CN LS-F25946-1) and of brain-derived neurotrophic factor (BDNF, Promega, CN G7610) were determined. All analyses were carried out using enzyme-linked immunosorbent assay (ELISA), according to the manufacturers’ instructions. Each sample was measured in duplicate, and the results are presented in pg/mL.

### 4.8. Statistical Analysis

The raw data were evaluated using the one-way analysis of variance (ANOVA) followed by Tukey’s post-hoc test (GraphPad Prism 6 Software). All results are presented as the mean ± standard error of the mean (S.E.M), and *p* < 0.05 was considered as statistically significant with 95% confidence.

## 5. Conclusions

The presented study is the first to provide data from in vivo experiments indicating the possibility of using O-1602 (GPR55 and GPR18 receptor agonists) in the treatment of DO/OAB. The following results deserve particular attention: (1) O-1602 did not affect the cystometric parameters in normal rats; (2) O-1602 alleviates/reverses the changes in cystometric and biochemical parameter characteristic of DO/OAB; (3) O-1602 affects the storage phase without impairing the voiding phase in rats with DO induced by RA; (4) the use of O-1602 can improve DO without affecting HR, MAP, or UP.

It should be emphasized that this is the first report indicating that O-1602 has a therapeutic potential in an animal DO model; (5) administration of O-1602 reverses in vivo symptoms of overactive bladder via inhibition of VAChTA, OCT_3_, and CGRP. However, the possibility of its use in patients with DO/OAB must be confirmed in clinical studies.

## Figures and Tables

**Figure 1 molecules-25-01384-f001:**
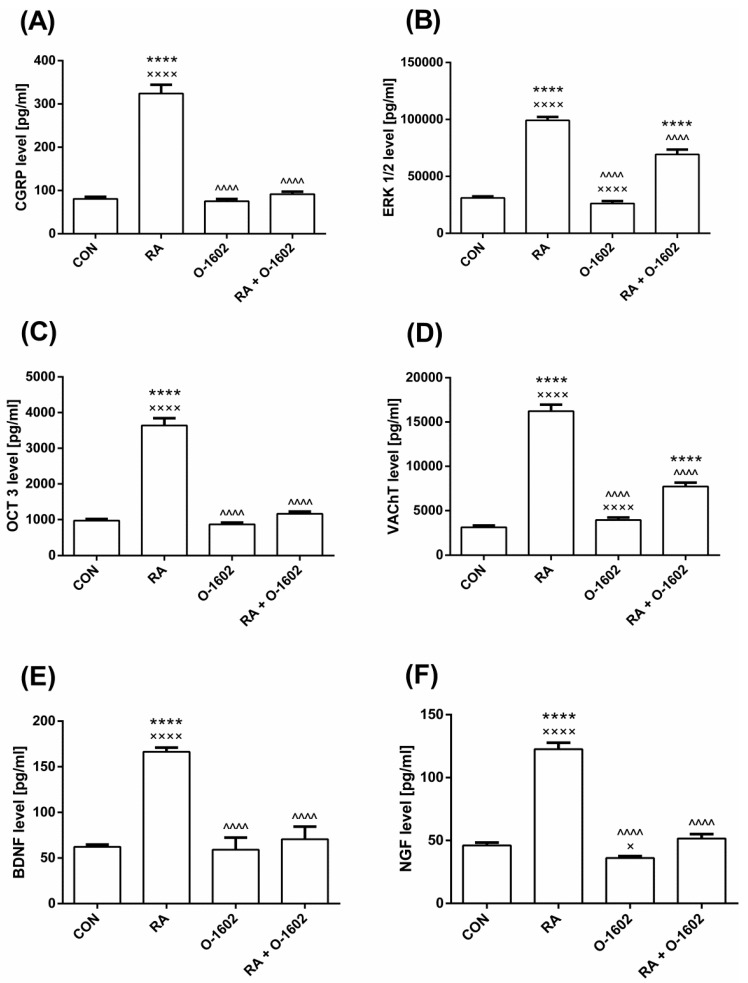
The influence of O-1602 administration in female rats with detrusor overactivity induced by RA on the level of (**A**) CGRP, (**B**) ERK1/2, (**C**) OCT_3_ in the bladder urothelium, (**D**) VAChT in the bladder detrusor muscle, (**E**) BDNF, and (**F**) NGF in the urine. One-way ANOVA: F(3,56) = 119.1, *p* < 0.0001 for CGRP; F(3,56) = 136.8, *p* < 0.0001 for ERK1/2; F(3,56) = 143.5, *p* < 0.0001 for OTC_3_; F(3,56) = 164.3, *p* < 0.0001 for VAChT; F(3,56) = 200.4, *p* < 0.0001 for BDNF; F(3,56) = 136.4, *p* < 0.0001 for NGF. All results are presented as the means ± SEM (*n* = 15 rats per group). The obtained data were assessed by the one-way ANOVA followed by Tukey’s *post hoc* test. *, ^, or ×: *p* < 0.05; **, ^^, or ××: *p* < 0.01; ***, ^^^, or ×××: *p* < 0.001; ****, ^^^^, or ××××: *p* < 0.0001. *: Significantly different from the CON; ^: Significantly different from the RA group; ×: Significantly different from the RA + O-1602 group. Abbreviations: BDNF—brain-derived neurotrophic factor; CGRP—calcitonin gene related peptide; ERK1/2– extracellular signal-regulated kinase 1/2; NGF—nerve growth factor; O-1602—agonist of GPR55 and GPR18 cannabinoid receptors; OCT_3_—organic cation transporter 3; RA—retinyl acetate; VAChT—vesicular acetylcholine transporter.

**Table 1 molecules-25-01384-t001:** The influence of O-1602 administration on the cystometric parameters in female rats with detrusor overactivity induced by retinyl acetate (RA).

Cystometric Parameters	CON	RA		O-1602		RA + O-1602	
**ANVC** [cmH_2_O]	2.221 ± 0.053	4.870± 0.442	**** ↑ ×××× ↑	2.207 ± 0.064	^^^^ ↓	3.027 ± 0.203	^^^^ ↓
**AUC** [cmH_2_O/sec]	13.60 ± 0.486	18.67± 0.785	**** ↑ ×× ↑	11.33 ± 0.522	^^^^ ↓ ×××× ↓	15.40 ± 0.600	^^ ↓
**BC** [ml/cmH_2_O]	0.2007 ± 0.006	0.1503± 0.006	**** ↓ ×××× ↓	0.2107 ± 0.009	^^^^ ↑	0.1948 ± 0.005	^^^^ ↑
**BCD** [sec]	27.12± 1.015	26.18± 1.186	ns	27.24 ± 1.126	ns	30.58 ± 1.847	ns
**BP** [cmH_2_O]	2.667± 0.211	4.547± 0.243	**** ↑ ×××× ↑	2.373 ± 0.124	^^^^ ↓	3.033 ± 0.215	^^^^ ↓
**DOI** [ml/cmH_2_O]	71.33± 5.275	177.5± 10.24	**** ↑ ×××× ↑	63.20 ± 5.183	^^^^ ↓ ×× ↓	100.5 ± 5.788	* ↑ ^^^^ ↓
**FNVC** [times/filling phase]	0.5067 ± 0.056	5.855 ± 0.484	**** ↑ ×××× ↑	0.2700 ± 0.038	^^^^ ↓	1.258 ± 0.240	^^^^ ↓
**ICI** [sec]	932.5 ± 31.87	647.6 ± 32.57	**** ↓ ×××× ↓	860.9 ± 38.33	^^ ↑	912.4 ± 50.08	^^^^ ↑
**MVP** [cmH_2_O]	36.93 ± 2.281	32.51 ± 1.945	ns	39.85 ± 2.258	ns	33.66 ± 2.270	ns
**PVR** [ml]	0.0713 ± 0.006	0.0687 ± 0.005	ns	0.0614 ± 0.006	ns	0.0748 ± 0.004	ns
**RT** [sec]	18.66 ± 0.625	19.97 ± 0.778	ns	21.17 ± 0.646	ns	20.49 ± 0.910	ns
**TP** [cmH_2_O]	7.040 ± 0.309	4.960 ± 0.291	*** ↓ ×××× ↓	6.653 ± 0.270	^^ ↑	7.507 ± 0.446	^^^^ ↑
**VE** [%]	88.13 ± 2.019	89.53 ± 1.978	ns	89.07 ± 2.050	ns	92.00 ± 1.100	ns
**VT** [ml]	0.7460 ± 0.036	0.4953 ± 0.026	** ↓ ×××× ↓	0.8740 ± 0.034	^^^^ ↑	0.8233 ± 0.070	^^^^ ↑
**VTNVC** [%]	63.11 ± 2.845	29.93 ± 1.359	**** ↓ ×××× ↓	68.67 ± 4.527	^^^^ ↑	58.10 ± 3.978	^^^^ ↑
**VV** [ml]	0.9443 ± 0.046	0.5237 ± 0.035	*** ↓ ×× ↓	0.7754 ± 0.068	^ ↑	0.8706 ± 0.098	^^ ↑

One-way ANOVA: F(3,56) = 25.80, *p* < 0.0001 for ANVC; F(3,56) = 25.82, *p* < 0.0001 for AUC; F(3,56) = 16.61, *p* < 0.0001 for BC; F(3,56) = 2.084, *p* = 0.1126 for BCD; F(3,56) = 22.62, *p* < 0.0001 for BP; F(3,56) = 56.21, *p* < 0.0001 for DOI; F(3,56) = 92.83, *p* < 0.0001 for FNVC; F(3,56) = 11.29, *p* < 0.0001 for ICI; F(3,56) = 2.293, *p* = 0.0879 for MVP; F(3,56) = 1.153, *p* = 0.3360 for PVR; F(3,56) = 2.020, *p* = 0.1215 for RT; F(3,56) = 10.91, *p* < 0.0001 for TP; F(3,56) = 0.8130, *p* = 0.4920 for VE; F(3,56) = 14.07, *p* < 0.0001 for VT; F(3,56) = 25.66, *p* < 0.0001 for VTNVC; F(3,56) = 7.659, *p* = 0.0002 for VV. All results are presented as the means ± SEM (*n* = 15 rats per group). The obtained data were assessed by the one-way ANOVA followed by Tukey’s post hoc test. *, ^, or ×: *p* < 0.05; **, ^^, or ××: *p* < 0.01; ***, ^^^, or ×××: *p* < 0.001, ****, ^^^^, or ××××: *p* < 0.0001. *: Significantly different from the CON; ^: Significantly different from the RA group; ×: Significantly different from the RA + O-1602 group. Abbreviations: ANVC—nonvoiding contractions amplitude; AP—arterial pressure; AUC—area under the pressure curve; BC—bladder compliance; BCD—bladder contraction duration; BP—basal pressure; CON— control group; DBP—diastolic blood pressure; DO—detrusor overactivity; DOI—detrusor overactivity index; FNVC—nonvoiding contractions frequency; HR—heart rate; ICI—intercontraction interval; MBP—mean blood pressure; MVP—micturition voiding pressure; NEB—nebivolol hydrochloride; O-1602—agonist of GPR55 and GPR18 cannabinoid receptors; OAB—overactive bladder syndrome; PVR—post-void residual; RA—retinyl acetate; RT—relaxation time; SBP—systolic blood pressure; TP—threshold pressure; UP—urine production; VE—voiding efficiency; VT—volume threshold; VTNVC—volume threshold to elicit nonvoiding contractions; VV—voided volume.

**Table 2 molecules-25-01384-t002:** The measurement of urine production, heart rate, and arterial pressure parameters.

Diuresis and Cardiovascular Parameters	CON	RA		O-1602		RA + O-1602	
**UP** [mL/day]	18.61 ± 0.52	17.69 ± 0.76	ns	20.07 ± 0.70	^ ↑ ×× ↑	16.99 ± 0.51	ns
**HR** [beats/min]	307.7 ± 8.574	318.8 ± 8.705	ns	295.5 ± 11.00	ns	290.9 ± 8.264	ns
**MAP** [mmHg]	104.2 ± 4.189	94.53 ± 2.888	× ↑	110.4 ± 3.399	^ ↑ ×× ↑	91.27 ± 3.939	ns

One-way ANOVA: F(3,56) = 4.417, *p* = 0.0074 for UP; F(3,56) = 1.868, *p* = 0.1455 for HR; F(3,56) = 5.839, *p* = 0.0015 for MAP. All results are presented as the means ± SEM (*n* = 15 rats per group). The obtained data were assessed by the one-way ANOVA followed by Tukey’s post hoc test. ^ or ×: *p* < 0.05; ××: *p* < 0.01; ^: Significantly different from the RA group; ×: Significantly different from the RA + O-1602. Abbreviations: MAP—mean arterial pressure; O-1602—agonist of GPR55 and GPR18 cannabinoid receptors; RA—retinyl acetate; UP—urine production.

**Table 3 molecules-25-01384-t003:** Biochemical study parameters.

Biochemical Parameters		CON	RA		O-1602		RA + O-1602		Comments
In the bladder urothelium	**CGRP** [pg/mL]	80.47 ± 4.967	324.3 ± 20.18	**** ↑ ×××× ↑	75.07 ± 5.330	^^^^ ↓	91.60 ± 5.715	^^^^ ↓	🞍 CGRP is one of the bladder’s afferent activity markers 🞍 in DO the CGRP level ↑
	**ERK1/2** [pg/mL]	31035 ± 1423	99125 ± 3118	**** ↑ ×××× ↑	26154 ± 2139	^^^^ ↓ ×××× ↓	69255 ± 4275	**** ↑ ^^^^ ↓	🞍 ERK 1/2 plays an important role in the control of detrusor muscle tone and bladder reflex activity 🞍 in DO induced by pain stimuli the ERK 1/2 level ↑
	**OCT_3_** [pg/mL]	971.7 ± 46.26	3640 ± 201.8	**** ↑ ×××× ↑	870.1 ± 47.27	^^^^ ↓	1166 ± 61.68	^^^^ ↓	🞍 OTC_3_ is an organic cation transporter involved in the release of acetylcholine from non-neuronal cells, it occurs only in the bladder urothelium 🞍 in DO the OTC_3_ level ↑
In the bladder detrusor muscle	**VAChT** [pg/mL]	3133 ± 186.5	16220 ± 744.9	**** ↑ ×××× ↑	3947 ± 272.6	^^^^ ↓ ×××× ↓	7726 ± 425.3	**** ↑ ^^^^ ↓	🞍 VAChT is a vesicular acetylcholine transporter that releases it from non-neuronal cells, it occurs only in detrusor cells 🞍 in DO the VAChT level ↑
In the urine	**BDNF** [pg/mL]	62.20 ± 2.536	166.5 ± 4.685	**** ↑ ×××× ↑	59.00 ± 3.457	^^^^ ↓	70.60 ± 3.549	^^^^ ↓	🞍 BDNF and NGF belong to neurotrophins and are produced in the bladder’s urothelial and smooth muscle cells 🞍 in DO the BDNF and NGF level ↑
	**NGF** [pg/mL]	46.07 ± 2.292	122.6 ± 5.058	**** ↑ ×××× ↑	36.00 ± 1.555	^^^^ ↓ × ↓	51.53 ± 3.553	^^^^ ↓	

One-way ANOVA: F(3,56) = 119.1, *p* < 0.0001 for CGRP; F(3,56) = 136.8, *p* < 0.0001 for ERK1/2; F(3,56) = 143.5, *p* < 0.0001 for OTC_3_; F(3,56) = 164.3, *p* < 0.0001 for VAChT; F(3,56) = 200.4, *p* < 0.0001 for BDNF; F(3,56) = 136.4, *p* < 0.0001 for NGF. All results are presented as the means ± SEM (*n* = 15 rats per group). The obtained data were assessed by the one-way ANOVA followed by Tukey’s *post hoc* test. *, ^, or ×: *p* < 0.05; **, ^^, or ××: *p* < 0.01; ***, ^^^, or ×××: *p* < 0.001; ****, ^^^^, or ××××: *p* < 0.0001. *: Significantly different from the CON; ^: Significantly different from the RA group; ×: Significantly different from the RA + O-1602 group. Abbreviations: BDNF—brain-derived neurotrophic factor; CGRP—calcitonin gene related peptide; ERK1/2– extracellular signal-regulated kinase 1/2; NGF—nerve growth factor; O-1602—agonist of GPR55 and GPR18 cannabinoid receptors; OCT_3_—organic cation transporter 3; RA—retinyl acetate; VAChT—vesicular acetylcholine transporter.

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
