# Peer review of "A Novel Alternative in the Treatment of Detrusor Overactivity? In Vivo Activity of O-1602, the Newly Synthesized Agonist of GPR55 and GPR18 Cannabinoid Receptors"

_molecules, 2020, doi:10.3390/molecules25061384_

Round 1
Reviewer 1 Report
Overactive bladder (OAB) is a common medical problem affecting a substantial population of males and females worldwide. Current therapeutic treatment options available for the treatment of OAB promote untoward side-effects. The present paper discusses a novel therapeutic approach in the treatment of OAB.
- Line 20: remove “an”
- Line 26: consider changing “might be” to stronger language, such as “may serve as”
- Lines 35-36: rephrase “is as often diagnosed in…” to “is diagnosed as often in…”
- Lines 50-51: consider changing “at least several weeks” to “several weeks at minimum”.
- Lines 67-68: consider rephrasing “on the functioning of both laboratory animals’ and humans’ bladders” to “bladder function in laboratory animals and humans”.
- Line 79: consider rephrasing “act only post-synaptically” to “exclusively act post-synaptically”
- Line 180: Change “inhibited in-vivo excessive bladder contractility” to “inhibited excessive bladder contractility in vivo”
- Lines 187-190 Consider re-wording (“Modifications to the cystometric parameters noted after the administration of the GPR55 agonist, i.e., increase in BC, BCD, ICI, TP, VT, VV and VTNVC and decrease in ANVC, AUC, BP, DOI and FNVC, prove that O-1602 can become an interesting alternative in pharmacological 189 treatment, both OAB-wet and OAB-dry.”)
- Lines 193-195: Reword or revise with correct punctuation: “Importantly, the lack of O-1602′s impact on MVP, PVR, RT, and VE indicates that in the applied dose it only affects the urine accumulation phase, without affecting the voiding phase” to either “Importantly, the lack of O-1602′s impact on MVP, PVR, RT, and VE indicates that the applied dose only affects the urine accumulation phase, without affecting the voiding phase”.
- Lines 204-206: Run-on sentence. Consider revising.
- Line 263: insert a space between GRP55 and receptor agonist.
- Line 295: change “an” to “a”.
- Line 302: Add a comma after “Marentette et al.”
Additional comments:
Consider adding a table summarizing the function/significance of each marker that was tested. The acronyms used become confusing and are hard to follow. A table at the beginning of the article to reference would enhance the quality of the manuscript.
Great job on the conclusion summary.
Author Response
Overactive bladder (OAB) is a common medical problem affecting a substantial population of males and females worldwide. Current therapeutic treatment options available for the treatment of OAB promote untoward side-effects. The present paper discusses a novel therapeutic approach in the treatment of OAB.
Line 20: remove “an”
Line 26: consider changing “might be” to stronger language, such as “may serve as”
Lines 35-36: rephrase “is as often diagnosed in…” to “is diagnosed as often in…”
Lines 50-51: consider changing “at least several weeks” to “several weeks at minimum”.
Lines 67-68: consider rephrasing “on the functioning of both laboratory animals’ and humans’ bladders” to “bladder function in laboratory animals and humans”.
Line 79: consider rephrasing “act only post-synaptically” to “exclusively act post-synaptically”
Line 180: Change “inhibited in-vivo excessive bladder contractility” to “inhibited excessive bladder contractility in vivo”
Lines 187-190 Consider re-wording (“Modifications to the cystometric parameters noted after the administration of the GPR55 agonist, i.e., increase in BC, BCD, ICI, TP, VT, VV and VTNVC and decrease in ANVC, AUC, BP, DOI and FNVC, prove that O-1602 can become an interesting alternative in pharmacological 189 treatment, both OAB-wet and OAB-dry.”)
Lines 193-195: Reword or revise with correct punctuation: “Importantly, the lack of O-1602′s impact on MVP, PVR, RT, and VE indicates that in the applied dose it only affects the urine accumulation phase, without affecting the voiding phase” to either “Importantly, the lack of O-1602′s impact on MVP, PVR, RT, and VE indicates that the applied dose only affects the urine accumulation phase, without affecting the voiding phase”.
Lines 204-206: Run-on sentence. Consider revising.
Line 263: insert a space between GRP55 and receptor agonist.
Line 295: change “an” to “a”.
Line 302: Add a comma after “Marentette et al.”
Response 1: According to the suggestions mentioned above, the required changes have been made in the manuscript. The manuscript was also checked by a native speaker
Additional comments:
Consider adding a table summarizing the function/significance of each marker that was tested. The acronyms used become confusing and are hard to follow. A table at the beginning of the article to reference would enhance the quality of the manuscript.
Response 2: As suggested by the Reviewer we added information about function/significance of each tested marker in Table 3. This will facilitate the reader easier analysis of the results contained in this table.
At the beginning of the manuscript we have added “List of abbreviations”. We hope that this will enhance the quality of the manuscript.
Great job on the conclusion summary.
Responce 3. Thank you very much for Your comments/suggestions and opinion.

Reviewer 2 Report
The authors investigate O-1602 as a therapeutic for detrusor overactivity.
Comments:
- O-1602 also is an agonist for GPR18. This needs to be include throughout the whole manuscript
- Introduction: why are muscarinic receptor antagonists prescribed
- Paragraph starting at line 53, based on this, what is the justification for looking at cannabinoid receptors? What limitations are there with the other targets?
- Line 73-what is meant by "is new"? Novel identification, novel determination that it is modulated by endocannabinoids?
- The introduction does not justify the use of GPR55 antagonist specifically? why not use a drug that modifies CB1 and CB2?
- Methods: were data normally distributed?
- Why this dose of O-1602?
- In the results (especially lines 101 onwards, there are a lot of terms not defined)
- Throughout the results you should specify that it is female rats
- For the biochemical studies it is not clear what was measured ie eg rna , protein, or via what method, western blot, elisa?
- The discussion requires a better framework: what was your hypothesis and does your data support this?
- Is there a conclusion?
Author Response
The authors investigate O-1602 as a therapeutic for detrusor overactivity.
Comments:
O-1602 also is an agonist for GPR18. This needs to be include throughout the whole manuscript.
Responce 1: Yes, O-1602 also is an agonist of GPR18 receptors. We added that information in the Introduction and Discussion section. In the manuscript, we have not focused on the effect of O-1602 on GPR18 receptors, since their location suggests that their stimulation does not affect bladder function. We included this information throughout the whole manuscript.
Introduction: why are muscarinic receptor antagonists prescribed
Responce 2: We've added information about antimuscarinics’ mechanism of action and their effects on the bladder in the Introduction section.
Paragraph starting at line 53, based on this, what is the justification for looking at cannabinoid receptors? What limitations are there with the other targets?
Responce 3: The justification for looking at cannabinoid receptors and limitations of the other targets have been mentioned in the Introduction section.
Line 73-what is meant by "is new"? Novel identification, novel determination that it is modulated by endocannabinoids?
Responce 4: We meant that the GPR55 receptors have been relatively recently identified and not yet well-known and well-studied. This has been clarified in the manuscript.
The introduction does not justify the use of GPR55 antagonist specifically? why not use a drug that modifies CB1 and CB2?
Responce 5: There are few reports on the possibility of using substances that modulate cannabinoid receptor activity in the therapy of patients suffering from OAB. Usefulness in OAB/DO therapy of substances that modify the activity of CB1 and CB2 receptors has already been tested [69, 71, 74, 84]. In the literature neither pre-clinical nor clinical studies that evaluated the possibility of using GPR55 receptor agonists in patients with DO were found. Therefore, the objective of the current research was to evaluate the effects of O-1602 on urodynamic parameters in the animal model of DO induced with the transient intravesical infusion of retinyl acetate in conscious female rats during cystometry.
This information has been included in the Introduction section
Methods: were data normally distributed?
Responce 6: Yes, outcomes were normally distributed.
Why this dose of O-1602?
Responce 7: The dose of O-1602 was selected on the basis of the literature data, and were confirmed/adjusted in their laboratory in preliminary studies [59]. The lowest effective dose of O-1602 (i.e. 0.25 mg/kg) was selected for the appropriate reaserch.
In the results (especially lines 101 onwards, there are a lot of terms not defined)
Responce 8: We have defined all of terms in the Results section. At the beginning of the manuscript we have added “List of abbreviations”.
Throughout the results you should specify that it is female rats.
Responce 9: We have specified that results koncern female rats.
For the biochemical studies it is not clear what was measured ie eg rna , protein, or via what method, western blot, elisa?
Responce 10: The levels of the biomarkers were measured in plasma and tissue homogenates using enzyme-linked immunosorbent assay (ELISA), according to the manufacturer’s protocols.
This information has been included in the Materials and Methods section.
The discussion requires a better framework: what was your hypothesis and does your data support this?
Responce 11: To the authors’ best knowledge, this is the first study in which the effect was assessed of O-1602 receptor agonist on bladder function in an animal model of RA-induced DO. The aim of the study was to check in vivo the effect of O-1602 on the micturition cycle of conscious animals female rats in the animal model of DO [63] previously developed by the research team and its applicability in DO/OAB therapy. Previously, no attempt had been made to determine the level of the OAB biomarkers after the administration of this agent. The results of cystometric and biochemical studies presented in this manuscript support our hypothesis.
This information has been presented in the Discussion section as the first paragraph.
Is there a conclusion?
Responce 12: The Conclusion section is in point 5 and is as follows:
The presented study is the first to provide data from in-vivo experiments indicating the possibility of using O-1602 (GPR55 and GPR18 receptor agonists) in the treatment of DO/OAB. The following results deserve particular attention: (1) O-1602 did not affect the cystometric parameters in normal rats; (2) O-1602 alleviates/reverses the changes in cystometric and biochemical parameters characteristic of DO/OAB; (3) O-1602 affects the storage phase without impairing the voiding phase in rats with DO induced by RA; (4) the use of O-1602 can improve DO without affecting HR, MAP, or UP.
It should be emphasised that this is the first report indicating that O-1602 has therapeutic potential in an animal DO model, (5) administration of O-1602 reverses in vivo symptoms of overactive bladder via inhibition of VAChTA, OCT3 and CGRP. However, the possibility of its use in patients with DO/OAB must be confirmed in clinical studies.
